# Towards a Topographically-Accurate Reflection Point Prediction Algorithm for Operational Spaceborne GNSS Reflectometry—Development and Verification †

**Lucinda King** [1,*], **Martin Unwin** [2], **Jonathan Rawlinson** [2], **Raffaella Guida** [1] and **Craig Underwood** [1]

1   Surrey Space Centre, University of Surrey, Guildford GU2 7HX, UK; r.guida@surrey.ac.uk (R.G.);
    c.underwood@surrey.ac.uk (C.U.)
2   Surrey Satellite Technology Ltd., Guildford GU2 7YE, UK; m.unwin@sstl.co.uk (M.U.);
    j.rawlinson@sstl.co.uk (J.R.)
*   Correspondence: l.s.king@surrey.ac.uk
†   This paper is an extended version of our paper published in IGARSS 2020, 26 September–2 October 2020.

**Abstract:** GNSS Reflectometry (GNSS-R), a method of remote sensing using the reflections from satellite navigation systems, was initially envisaged for ocean wind speed sensing. In recent times there has been significant interest in the use of GNSS-R for sensing land parameters such as soil moisture, which has been identified as an Essential Climate Variable (ECV). Monitoring objectives for ECVs set by the Global Climate Observing System (GCOS) organisation include a reduction in data gaps from spaceborne sources. GNSS-R can be implemented on small, relatively cheap platforms and can enable the launch of constellations, thus reducing such data gaps in these important datasets. However in order to realise operational land sensing with GNSS-R, adaptations are required to existing instrumentation. Spaceborne GNSS-R requires the reflection points to be predicted in advance, and for land sensing this means the effect of topography must be considered. This paper presents an algorithm for on-board prediction of reflection points over the land, allowing generation of DDMs on-board as well as compression and calibration. The algorithm is tested using real satellite data from TechDemoSat-1 in a software receiver with on-board constraints being considered. Three different resolutions of Digital Elevation Model are compared. The algorithm is shown to perform better against the operational requirements of sensing land parameters than existing methods and is ready to proceed to flight testing.

**Keywords:** GNSS-R; topography; data compression; on-board data processing

## 1. Introduction

GNSS reflectometry (GNSS-R) is a method of remote sensing which uses navigation signals such as those from GPS as "Signals of Opportunity" in a system similar to bistatic radar. The navigation satellites transmit L-band signals which are incident all over the Earth, that interact with and reflect from the surface (thereby acquiring information about the geometric, textural and dielectric properties of the surface) and are collected by a receiver, in this context a low Earth orbit spacecraft. The method has to-date most typically been used for sensing ocean wind-speed [1], but recent work has focused on the possibility of using the method to target land parameters such as soil moisture (e.g., [2,3]).

The primary observable of GNSS-R is the delay-Doppler map (DDM) which is built up by correlating the reflected signal with another copy (either the direct signal or a local replica—in this paper the latter is considered) shifted to have a range of different delays and Doppler values, thereby building up a map of reflected power centred around the delay and Doppler of the reflection point. This point is the specular point (SP) for forward-scattering GNSS-R, which is the most common form currently, although the issues addressed herein also apply to backscattering mode. As the reflected signals are very weak, the reflection

points must be predicted in advance and tracked using an open loop method to direct the correlators to look for the reflected signal in the appropriate delay and Doppler search space [4].

The prediction of the reflection points requires the positions of the transmitter (the GNSS satellite), the receiver (the spacecraft in LEO receiving reflections—in this paper data from TechDemoSat-1 (TDS-1) is used [5])—and a model of the Earth's surface. Current GNSS-R spaceborne instrumentation uses a quasi-spherical Earth approximation, in which the Earth model used is the WGS-84 ellipsoid [6]. The quasi-spherical algorithm works by first scaling the Earth ellipsoid to a unit sphere, and applying the same transformation to the receiver and transmitter positions. The specular point is then determined in the transformed co-ordinates, using a method described in [7], before being scaled back to the ellipsoid. A full description of the method, including comparison with other methods and justification for its use on current instrumentation, can be found here [4].

The quasi-spherical method does not account for topography, but it has been sufficient for ocean scatterometry applications because the ocean surface approximately follows the geoid, with deviations less than ∼100 m [8], and is unaffected by topography (see Figure 1a). For land sensing such an approximation will not always be sufficient due to the impact of terrestrial topography, which could result in inaccurate prediction of the reflection point, as shown in Figure 2. The centre of the DDM, or "cross-hairs", of both DDMs in Figure 1 corresponds to the predicted specular reflection point (using the quasi-spherical method). It should be noted that the quasi-spherical method does have its own errors in prediction of the specular point, even over the ocean, which have been shown in some cases to impact the inversion of the DDMs to ocean wind-speed values due to an asymmetry in power distribution in the bins around the SP [9]. As inversion mechanisms for land parameters typically use the peak reflectivity (derived from the peak power bin only) the impact of these errors is lessened but they should still be considered in future analysis, as any errors from the initial smooth-Earth estimate will be incorporated into the topographically-corrected result.

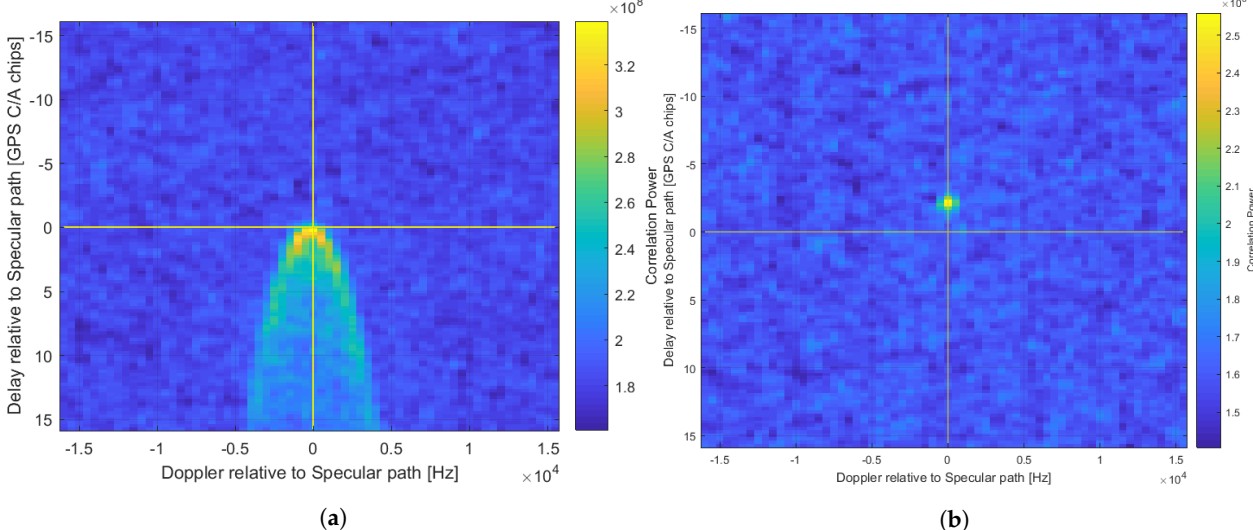

(a) (b)

**Figure 1.** Example DDMs from different Earth surfaces. (**a**) A typical DDM collected over the ocean, with the characteristic horseshoe shape showing the spreading of power over the surface. The peak power is located on the DDM cross-hairs as the specular point has been predicted with sufficient accuracy by the quasi-spherical model. (**b**) A DDM from a dataset over low-elevation mountains in South Sudan (approximate elevation 500 m). The cross-hairs represent the predicted reflection point, showing the effect of topography on the location of the peak reflected power in the DDM.

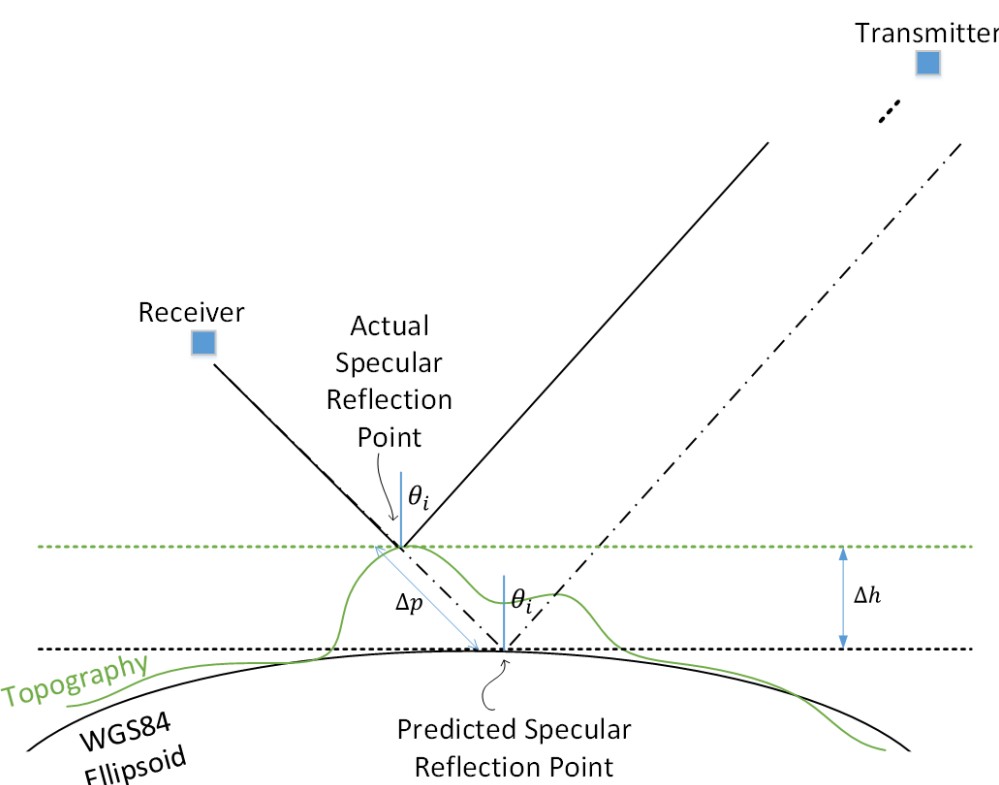

**Figure 2.** Diagram with simplified geometry demonstrating the influence of topography on specular reflection point prediction. The transmitter is considered to be far enough removed (20,200 km altitude for GPS) that the incoming direction of the incident signals at two nearby points can be assumed parallel. This diagram assumes that the elevated surface is flat, which is not often the case, but is a simplification to allow the most basic form of the algorithm to be implemented (see Section 2). Reproduced from [10] with permission.

This paper presents an algorithm for predicting GNSS-R reflection points in the presence of topography, which has been specifically designed for operational use on-board a small satellite. A recent paper by Gleason [11] has introduced a similar specular point prediction algorithm, developed independently, which includes the addition of a surface height term; however this is stated to only account for "low-lying land" and an operational limit is not given. In addition the algorithm in question was developed for a software receiver and although the paper mentions the possibility of its use on-board a reflectometry instrument, the actual implementation is not discussed. Other recent works include a method for modelling the effect of topography on DDMs [12], and a post-processing algorithm for accurately geolocating the reflection points in the presence of topography [13]. Whilst both these methods are valuable, they are designed for use on the ground. The objective of this paper is to introduce an algorithm that can work on-board, enabling reflected power to be captured by a GNSS-R instrument no matter the elevation of the reflecting surface—which is not currently possible. Should more accurate geolocation be required, this can be achieved on the ground using algorithms such as those referenced previously. In this way the algorithm presented here and the previously proposed post-processing algorithms are complementary.

The structure of this paper is as follows: in Section 2 there is a discussion of the motivation for using GNSS-R for sensing land parameters and the associated need to correct for topography when predicting reflection points. Section 3 contains a description of the proposed topographically-accurate reflection point prediction algorithm (TARPP) and the Digital Elevation Model chosen. The test procedures used to analyse the algorithm performance are given in Section 4 and results of these tests, with discussion, are presented in Section 5. The paper finishes with a conclusion in Section 7.

The study presented in this paper is a continuation of work introduced at the IGARSS conference in 2020 [10].

## 2. Motivation and Requirements

Land parameters such as soil moisture and Above Ground Biomass (AGB) have been designated as Essential Climate Variables (ECVs) by an international organisation, the Global Climate Observing System (GCOS) [14,15]. This confirms that measurement of these variables is critical for monitoring and modelling climate change. In addition, soil moisture, for example, is important for forecasting local weather and flood risks [16].

GCOS has identified monitoring objectives for the ECVs which include a call to reduce the spatial and temporal gaps in datasets collected from space [17]. GNSS-R is ideally placed to meet this challenge. The L-band signals are sensitive to parameters such as soil-moisture, freeze-thaw status and AGB [3,18–20]; and the low size, weight and power (SWaP) platforms used for the method enable—in a complementary way to other technologies—single satellites or even constellations to be built and launched more quickly, cheaply and flexibly than large traditional radar missions. GNSS-R technology must be adapted to optimally target land parameters and this includes developing means of accurately predicting the reflection points over the land.

An associated objective, which also requires accurate knowledge of the reflection points over the Earth's surface is that of data compression via windowing, which will also help improve instrumentation and support GNSS-R land sensing missions.

For both the objectives—land sensing over all elevations, and data compression—accurate prediction of the reflection points is required, for the following reasons. These are particularly relevant for downlink of on-board processed DDMs—rather than "raw" sampled IF (intermediate-frequency) data—as would be the case for an operational mission. These reasons can be translated into objectives for the topography algorithm, as shown in Table 1.

- The DDM is built up through correlation of the received reflected signals with a clean replica that is systematically shifted through the various combinations of delay and Doppler $(d, D)$ to calculate the received power at every pixel. If the prediction of $(d, D)$ of the specular reflection is correct, this reflected power will be present at the centre of the DDM. However, if the prediction is wrong, this power can appear at other positions in the DDM or sometimes may not appear at all (see Equation (3)). An example of the latter can be seen in Figure 1b, where the peak reflected power is offset from the centre due to topography. If the reflected power is not captured within the DDM at all then that measurement is lost.

- For forward-scattered DDMs, the centring of the reflected power allows the definition of a "Noise Box" in the null space at the top of the DDM, which corresponds to delays shorter than the predicted specular point (SP) delay—over the ocean such delays are typically not physically possible and so any power in this region can be assumed to be noise. This Noise Box is used to help measure noise power and thus calculate absolute received signal power from measured Signal-to-Noise Ratio (SNR). Reflections from unaccounted-for topography can contaminate the area resulting in inaccurate measurements or an inability to use the method at all [21].

- The minimum requirement for successful recording of reflected power on the space-craft is that the peak is located somewhere within the DDM (the effect on the Noise Box and thus calibration notwithstanding). If necessary, users can perform more accurate geo-location on the ground through re-processing. However, a stronger requirement is for the peak reflected power to be recorded as close to the centre of the DDM (delay offset = Doppler offset = 0) as possible at the on-board processing stage. This allows more efficient methods of data compression (i.e., windowing the DDM) to be used, which can be an enabler for usable data to be disseminated to users on the ground faster. This is key for soil moisture in particular. For certain missions e.g., those using very low power platforms, significant compression may be

essential to close the data downlink budget. In general, regardless of the application or size of the platform, enabling better data compression is desirable for more efficient operation. For example, to enable 24/7 Level 1b DDM downlink operations on DoT-1 (an SSTL spacecraft on which this algorithm will be tested) it has been calculated that DDMs should be reduced to a window with 64 pixels in area. An 8 × 8 box has been chosen for initial testing (over e.g., 16 × 4) as this places a stronger test on the path length prediction requirement, and will also capture more of the DDM width, which may help to record multiple peaks if they should occur in e.g., mountainous terrain. Figure 3 demonstrates the impact of this desired windowing on DDMs generated with and without a topographically accurate algorithm.

- Some mission concepts that have recently been developed have proposed a so-called "coherent channel" to track the peak reflected power with longer coherent integration times, which allows carrier phase information to be preserved. As this results in twice as much data (both I and Q components are stored), the channel should ideally monitor one pixel of the DDM only, and therefore the peak reflected power must be located accurately in this coherent pixel. An accurate topography algorithm is part of the process for achieving this.

**Table 1.** Spectrum of Objectives for Topography Algorithm. The stretch objective for coherent pixel tracking is not targeted by the TARPP v1 algorithm tested as part of this study, but is included to give the full context of the current work and future directions.

| Algorithm Outcome | Land Sensing Enabled Over Whole Globe | Noise Box Preserved | Data Compression Enabled (Windowing) | Coherent Pixel Tracking |
|---|---|---|---|---|
| **Baseline Outcome** Requirement: SP power captured within 128 × 20 pixel window | ✓ | Not guaranteed | × | × |
| **Good Outcome** Requirement: SP power captured within 8 × 8 pixel window | ✓ | ✓ | ✓ | × |
| **Best Outcome—"Stretch Objective"** Requirement: Location of SP power in DDM window known to 1 pixel. | ✓ | ✓ | ✓ | ✓ |

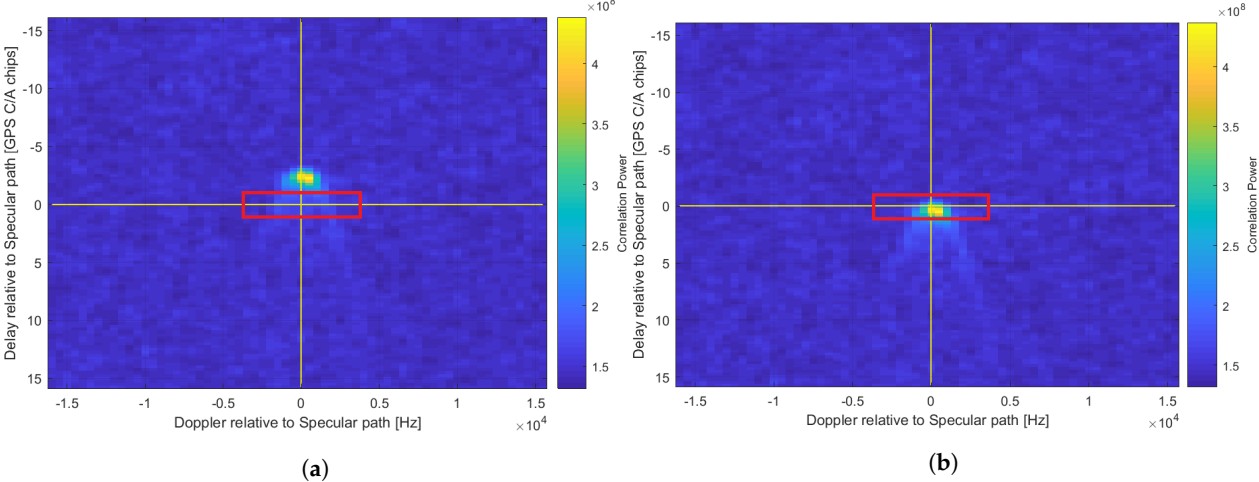

(**a**)  (**b**)

**Figure 3.** These DDMs demonstrate the impact of topography on the DDM and the impact of compressing the DDM using windowing. The red box represents an 8 × 8 pixel window. In this case, without the topography algorithm the peak power data would be lost as it falls outside the window. The surface elevation where this DDM was collected was approximately 400 m. (**a**) Land DDM generated with quasi-spherical algorithm. (**b**) The same DDM generated with the topography algorithm.

In a TechDemoSat-1 (TDS-1) DDM the centre, or cross-hairs, corresponds to the delay-Doppler $(d, D)$ value of the specular reflection point. The DDM is normalised so that this point has $d = D = 0$, and other delays and Dopplers are measured relative to this. In a TDS-1 Level 1b DDM the delay dimension extends over 128 pixels, equivalent to 16 chips (each chip is equivalent to 293 m, or 977 ns, for the GPS L1 C/A code) on either side of the zero-delay point, and 10 Doppler pixels each way. The raw TDS-1 data used in this study (discussed in Section 4) has been processed by a software receiver and has 64 Doppler pixels in total (slightly more than the L1b DDMs so more of the search space is visible for analysis- it does not effect the processing of the DDMs otherwise). These extend to approximately ±15 kHz relative to the specular point (see Figure 1b). Considering the delay dimension, if the path length of the reflection is predicted with an inaccuracy greater than 16 chips due to omitted topography, the reflected power will not appear in the DDM. Using a simplified view of the Earth's surface (not accounting for surface slopes), as shown in Figure 2, an elevation above the ellipsoid greater than ~2300 m is sufficient to produce these path length differences or greater. This is calculated using the following:

$$\frac{2\Delta h}{cos(\theta_i)} = \Delta p \tag{1}$$

$$16 \text{ chips} = 16 \times 293 \text{ m}$$
$$= 4688 \text{ m} \tag{2}$$

$$\Delta h > \frac{4688}{2} cos(\theta_i) \text{ m} \tag{3}$$

where $\Delta h$ is the elevation above the ellipsoid, $\theta_i$ is the incidence angle between reflection point and receiver and $\Delta p$ is the resulting path length change.
For the limiting case of $\theta_i = 0$ (nadir),

$$\Delta h > 2344 \text{ m} \tag{4}$$

For high incidence angles, smaller height deltas above the ellipsoid can result in the same path length change, which will increase the required prediction accuracy. For example, assuming 70° as the incidence angle limit means that elevations over 800 m will result in unacceptable path length changes, unless the elevation above the ellipsoid is accounted for.

According to hypsometric curves of the Earth's surface ([22]) ~10% of the terrestrial surface is over 2300 m elevation above sea level. This includes regions such as the Tibetan Plateau (average elevation over 4000 m), an area which has a well-documented impact on the Asian climate and weather and for which monitoring of soil moisture is therefore very important [23]. This is just one example of an area requiring denser spatio-temporal sampling from satellite measurements, which could be addressed by GNSS-R technology—but only if the topography requirements are addressed.

The percentage of surface potentially creating reflections outside of the DDM is substantially greater than 10% once data compression is considered. For example, in standard mode CYGNSS compresses the downlinked Level 1 DDMs produced on-board to 1 delay chip "above" and 3 "below" the nominal specular point (17 delay × 11 Doppler pixels) [24]. If 70° is again assumed as the incidence angle limit, then heights above 50 m will cause the reflections to fall outside the downlinked DDM. This corresponds to ~90% of the terrestrial surface.

It is acknowledged that the assumption that the elevated surface is flat, and that the effect of topography does not shift the reflection point laterally, is a simplification which could lead to inaccuracies in the specular point estimation. However, this assumption has been made for this algorithm as the intention is to operate on-board small satellites where non-iterative algorithms are preferred. The intention is to explore what can be achieved with a very simple algorithm, which will open up the possibilities for the implementation

of the technology. Future work (discussed in Section 6 will include development of more accurate algorithms including e.g., surface slopes, and the possibilities of multiple reflection points.

## 3. Algorithm

### 3.1. Baseline Algorithm (Elevation Only)—TARPP v1

The baseline algorithm that has been developed is described in Figure 4 and is referred to as TARPP v1—a Topographically Accurate Reflection Point Predictor (version 1). This algorithm is non-iterative, which is preferable for inclusion in flight software, and applies a height correction extracted from an on-board Digital Elevation Model (DEM) to an initial "smooth-Earth" estimate—"smooth" here meaning that the estimate uses a purely mathematical model of the Earth which does not include topography. In this case the initial estimate is generated using the existing quasi-spherical method, which is described elsewhere [4,21]. The algorithm refers to a "Reflection Point" rather than a "Specular Point" because although this study concerns forward-scattering GNSS-R, in general the algorithm could be applied to any predicted reflection point, including for example backscattered reflection points, which will be addressed in future work.

Highly accurate geolocation in both horizontal and vertical dimensions, such as for altimetry, is not the primary focus of the algorithm, however it will be ensured that future operational implementation of the algorithm will be transparent so any effects can be undone by users, should they wish to recalculate a more accurate reflection point on the ground.

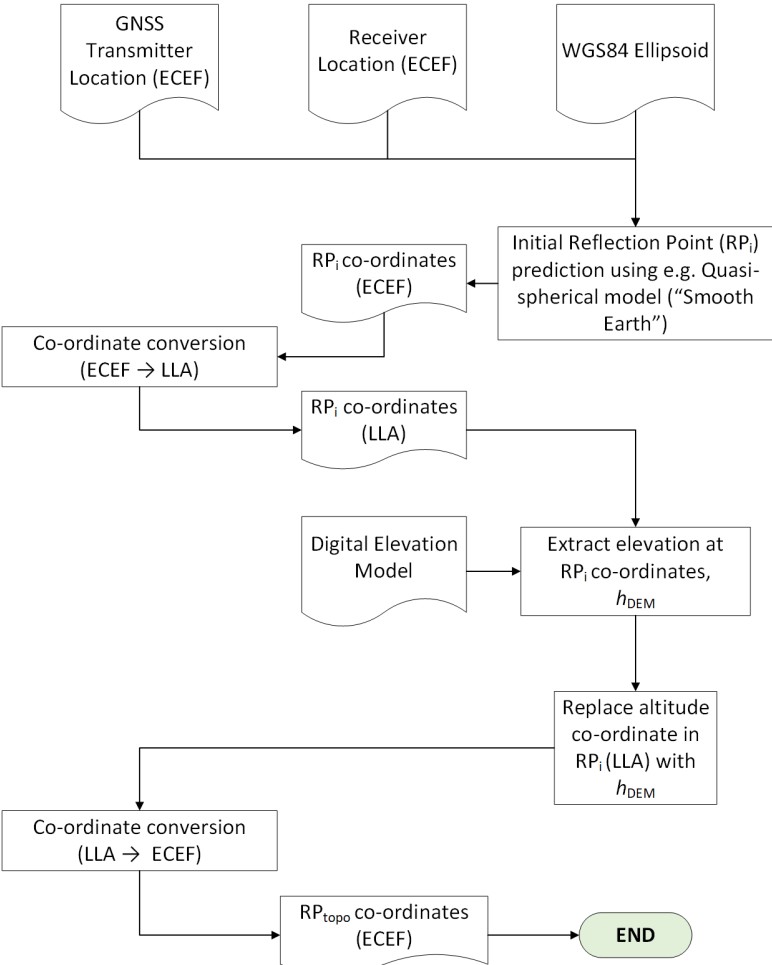

**Figure 4.** Diagram of the baseline version of the Topographically Accurate Reflection Point Prediction algorithm (TARPP v1).

The initial estimate is calculated in Earth-Centred-Earth-Fixed (ECEF) co-ordinates using the receiver and transmitter positions and the WGS84 ellipsoid. The point co-ordinates are then transformed into Latitude-Longitude-Altitude (LLA) format and used to index into a DEM and extract the elevation at that point. The altitude co-ordinate, originally zero, is then set to this elevation value and the point transformed back to ECEF.

The TARPP v1 algorithm has been developed in MATLAB as part of a software-defined GNSS-R receiver that processes raw TDS-1 data. It has undergone testing in this environment, the method and results of which are discussed in Sections 4 and 5 respectively.

### 3.2. Digital Elevation Models and On-Board Constraints

A key component of the algorithm is the Digital Elevation Model (DEM) from which the elevation is extracted, and therefore the choice of DEM should be evaluated carefully. Criteria to be considered include the resolution, extent over the Earth and data sources. The EarthEnv suite of topographic variables [25] provides several DEMs of various horizontal resolutions and aggregation methods. DEMs from this suite have been chosen for use as they are near-global in scope (84N - 56S), and the availability of other resolution DEMs and slope data within the suite which have been generated using the same source data will make comparison of different resolutions, and the development of the next stage of the algorithm, simple and consistent. The EarthEnv topographic variables, including the DEMs, use the 250 m GMTED2010 [26] as the primary source, with the 90 m SRTM4.1dev [27] used for validation. The accuracy of GMTED2010 is affected by the presence of vegetation, however this is non-trivial to resolve and beyond the scope of this study. Approximately 10.9 m of the global mean error of SRTM, on which GMTED is based, is due to vegetation [28], however in the worst case this would result in a few pixels of error in the reflection point location in the DDM. This means it would need to be accounted for to achieve the "stretch objective" of coherent pixel tracking shown in Table 1, but this is not a goal of the TARPP v1 algorithm and so is not considered further here.

There are different options available for the method used to aggregate the source data to the different spatial grains. The mean was chosen, rather than the minimum or maximum, to reduce the possibility that outliers would result in a significant over or under compensation of the elevation at a certain point. It should be noted that the EarthEnv DEM is referenced to the EGM96 geoid surface but the positions of the transmitter and receiver, which are calculated using GNSS solutions, are referenced to the WGS84 ellipsoid. As discussed in Section 1, the geoid and the ellipsoid do not deviate by more than ±100 m. Initial testing on a selected dataset in the Taklamakan desert region which showed a bias demonstrated that including a geoid correction in the DEM did not show an improvement to the bias in this case. Therefore due to this, and to expedite the analysis, the geoid undulations have not been included in the DEMs for the purposes of this study.

The operational constraints affecting the choice of DEM for a spaceborne platform are:

- Uplink rate, affecting file size.
- On-board storage, also affecting file size.
- Processing requirements for unpacking and using on-board.

The larger the file to be uploaded the more passes required, which can stretch over several days. For future missions using on-board topography algorithms, there will be the option to upload the DEM on the ground before launch, which mitigates this constraint. However there are several GNSS-R capable spacecraft currently in orbit that could be adapted to address the land sensing monitoring objectives and so consideration of uplink constraints is important. The file size limit will constrain the horizontal and vertical resolutions of the DEM chosen for use.

The other aspect of the resolution trade-off is the comparison of the horizontal resolution with the spatial resolution of GNSS-R scatterometry over the land. As discussed in recent literature [29,30], scattering over the land surface is a complex mix of coherent and incoherent scattering, affected by many variables such as surface roughness, vegetation cover and large-scale topography. Purely coherent scattering, with spatial resolution comparable

to the first Fresnel Zone (∼1 km) is possible, but is mostly limited to inland water bodies, a very small fraction of the land surface. Primarily, the scattering over the land surface has a large diffuse component and so has a lower spatial resolution. However, the horizontal resolution of the DEM should be chosen to correspond with the best possible case of spatial resolution, that of coherent scattering, so that in these cases the DEM resolution is not broader than the scattering resolution. This instantaneous spatial resolution, of the order of 1 km, is combined with the motion of the reflection point over the surface during the period of incoherent integration (in this case 1 s), which is dominated by the motion of the LEO receiver. Therefore the resolution is of the order of 7 km in the along-track dimension, and the chosen horizontal DEM resolution should be comparable with this.

Regarding the vertical resolution, if a 1 m level of precision is assumed (storing the values as 16 bit signed integers) this gives a maximum error in a stored value of 0.5 m. Using Equation (1) this gives a worst case error in the path length of 57 m. This is ∼1/5 of a chip which is acceptable with respect to the requirement to centre the reflected power in the DDM window. Storing the DEM values as integers significantly reduces the size of the DEM. The EarthEnv DEM values range from −412 m to 7919 m which can be adequately represented using 16 bit signed integers. There is no benefit to switching to unsigned integers as the range is more than large enough, and it would require adding a constant factor to every DEM value. Although this is trivial to implement in software, it is an unnecessary complication. On other platforms, if the available memory on-board is severely limited the DEM values could be represented by unsigned 8 bit integers and similarly converted using additive and multiplicative factors, however this would have an impact on the vertical accuracy and result in an increased path-length error.

This algorithm has testing planned on-board the SSTL DoT-1 technology demonstration satellite and so this spacecraft has been used as an example for the development and testing of the algorithm. DoT-1 is a 17.5 kg satellite in a 530 km polar orbit, with an S-band up and downlink. The uplink rate is the dominant operational constraint in this case, as the satellite is already in orbit. Flight software on DoT-1 is stored in non-volatile memory, with 128 MB available, and the DEM must fit within these bounds. The EarthEnv 5 km and 10 km resolution DEMs meet the on-board memory constraints, and so these will be chosen for test. The 1 km DEM will not feasibly fit on-board but has been selected for comparison. Therefore the DEM provisionally selected for use on-board is the 5 km resolution, and will be compared with the higher and lower resolution DEMs, 1 km and 10 km.

## 4. Test Procedure

### 4.1. Datasets

This study has made use of data from the TDS-1 satellite, which are generated in two forms on-board. Firstly, Level 1a DDMs, which are combined with metadata to form Level 1b DDMs which are then disseminated via the MERRByS online portal [31]. These are 128 by 20 pixels in delay and Doppler respectively—details of how the on-board instrument generated these can be found in [32]. Secondly, Level 0 "raw" data—sampled IF data which were not processed into DDMs on-board. A small number of sample Level 0 datasets are available from MERRByS, however this study had access to a larger catalogue of raw data.

The advantage of using raw data is that it allows simulation of on-board processing using real satellite data, with the ability to apply different processing methods to those used on-board currently. A disadvantage is that the results are dependent on the quality and characteristics of the real data.

There were approximately 60 datasets containing land or partial land reflections in the TDS-1 raw data catalogue, six of which were selected for intensive testing of the algorithm, as described in Table 2. Each dataset contained reflection tracks from several GNSS satellites (only GPS reflections have been considered in this study) and these are distinguished by their PRN number, which refers to the unique Pseudo-Random Noise code which modulates the carrier wave for that satellite [33].

**Table 2.** Raw TDS-1 datasets used in this study.

| Dataset Test ID | Collection Date and Timeslot | No. of DDMs | Approx. Location | Land Cover | PRN | Track Start | | | Track End | | |
|---|---|---|---|---|---|---|---|---|---|---|---|
| | | | | | | Co-Ordinate | Elevation (m) | El. angle at SP (°) | Co-Ordinate | Elevation (m) | El. angle at SP (°) |
| TT1 | 2015-01-11, H18 | 126 | Texas, USA | Open shrubland and grassland. Some mountainous regions. | 15 | 35.585, −97.421 | 316 | 65.3 | 30.627, −98.897 | 385 | 61.1 |
| | | | | | 18 | 35.650, −103.721 | 1259 | 59.7 | 30.596, −104.891 | 994 | 59.1 |
| | | | | | 21 | 36.702, −101.193 | 861 | 69.3 | 31.637, −102.525 | 849 | 64.3 |
| TT2 | 2015-06-05, H12 | 129 | Sahara Desert (Algeria to Mali) | Desert | 12 | 25.665, −2.526 | 298 | 66.1 | 18.709, −4.045 | 284 | 66.7 |
| | | | | | 15 | 24.408, 0.665 | 332 | 74.0 | 17.309, −1.029 | 288 | 79.0 |
| | | | | | 24 | 29.132, −0.741 | 342 | 56.0 | 22.345, −2.368 | 352 | 48.3 |
| TT3 | 2015-11-04, H06 | 130 | Far Eastern China (Taklamakan Desert) to N. India. | Desert and mountains | 5 | 41.255, 81.677 | 985 | 61.2 | 34.417, 79.510 | 5504 | 52.9 |
| | | | | | 20 | 37.967, 76.976 | 1522 | 67.3 | 30.982, 75.161 | 216 | 75.9 |
| | | | | | 29 | 40.964, 76.881 | 3910 | 69.0 | 33.818, 74.953 | 1631 | 64.1 |
| TT4 | 2015-11-19, H18 | 130 | Arizona to Californian coast, USA | Desert and shrubland, mountains (Sierra Nevada). Some tracks cross the ocean. | 15 | 36.069, −110.878 | 1713 | 53.1 | 29.163, −113.060 | 0 | 49.9 |
| | | | | | 18 | 35.530, −117.960 | 948 | 67.9 | 28.472, −119.648 | 0 | 70.9 |
| | | | | | 20 | 38.840, −111.737 | 2050 | 50.9 | 32.107, −113.652 | 388 | 42.9 |
| | | | | | 21 | 38.055, −116.500 | 2060 | 67.5 | 31.018, −118.364 | 0 | 60.0 |
| TT5 | 2018-10-06, H12 | 128 | South Sudan to DR Congo | Savanna and forests. Some rivers. | 8 | 1.589, 27.740 | 682 | 46.6 | −5.211, 26.360 | 584 | 54.2 |
| | | | | | 11 | 6.872, 27.991 | 466 | 75.9 | −0.195, 26.493 | 585 | 71.4 |
| | | | | | 18 | 7.140, 30.227 | 414 | 73.4 | 0.097, 28.772 | 1166 | 66.2 |
| | | | | | 23 | 6.338, 25.489 | 649 | 54.9 | −0.560, 24.021 | 458 | 55.0 |
| TT6 | 2018-10-11, H12 | 129 | DR Congo | Forest, some mountainous regions, savanna. Some lakes. | 8 | −7.152, 27.347 | 639 | 49.3 | −14.058, 25.878 | 1115 | 57.2 |
| | | | | | 11 | −2.067, 27.266 | 939 | 77.4 | −9.205, 25.726 | 795 | 71.7 |
| | | | | | 18 | −1.813, 29.517 | 2158 | 70.9 | −8.937, 28.073 | 1452 | 64.0 |

### 4.2. Method—MATLAB Testing

The aim of the algorithm is to centre the peak reflected power within the DDM window to ensure that the data is captured despite high elevation reflection surfaces and compression constraints. Therefore analysing the position of the peak power pixel relative to the centre of the DDM is a good test for the performance of the algorithm. Whilst it is accepted that in the case of diffuse scattering the peak of the DDM can lag in the delay axis behind the true position of the specular point [34], the testing will proceed with the peak power being used as a proxy for the specular point, as this is currently accurate enough for our purposes.

For testing purposes the TARPP v1 algorithm was implemented in MATLAB as a module of an existing GNSS-R software receiver. This software receiver is capable of processing raw TDS-1 data into DDMs using a variety of methods, including the quasi-spherical method currently used on-board (described in Section 1). The six test datasets were processed first using this method and then using the TARPP algorithm, with three different DEM horizontal resolutions −1, 5 and 10 km. A 5 km resolution was the provisional value chosen for on-board use as discussed in Section 3.2, and a resolution either side was chosen for comparison.

### 4.3. Analysis 1a—Peak Power Offset Graphs

The first analysis assessed the offset of the peak power from the centre of the DDM, using all pixels within ±10 Doppler pixels of the delay axis to simulate the current on-board DDM window (128 × 20 pixels). Within the test datasets there were several DDMs of reflections which had very weak coherent components and in these cases the DDMs had a noise-like quality and the highest power pixel was randomly distributed, as can be seen in the Results section. Preliminary work using a Gaussian filter to smooth the DDMs before running the analysis did not show an improvement in the results. At this stage a consistent method for automatically removing the noise-like DDMs has not been resolved and is still under development. Not all of the large delay offsets seen in the results are due to noise, as some coherent reflections can be seen at offsets away from the centre, even after the topography algorithm has been applied. This will be addressed further in the Results and Discussion section. This analysis was conducted on each test dataset separately.

### 4.4. Analysis 1b—Peak Power Offset Histograms

This analysis used histograms to display the distribution of the peak power offset from the centre. It was conducted on both the test datasets indivdually and a consolidated set of data from all PRNs from all datasets, to give an overall assessment of the algorithm performance.

### 4.5. Analysis 2—Compression Boxes

This analysis tested the ability of the algorithm to place the peak power pixel within various sized boxes around the centre of the DDM, simulating different window sizes that could be used for data compression. The box sizes tested were 4 × 4 pixels, 8 × 8 pixels and 20 × 20 pixels. The 8 × 8 pixel box could theoretically enable 24/7 operations on DoT-1, as discussed in Section 2. This analysis was conducted on each test dataset separately and the results were also combined to form an overall score for each DEM resolution for the consolidated dataset.

## 5. Results and Discussion

This section contains the test results of the TARPP v1 algorithm, using the analyses described in Section 4 on the datasets given in Table 2. Each section includes a comparison of the different DEM resolutions.

### 5.1. Analysis 1a

Analysis 1a presents the offset of the peak power from the DDM centre throughout the full DDM track. For conciseness, a graph for one PRN from each dataset has been selected for presentation, which is representative of the results from that test dataset. The results, as shown in Figure 5, show that the TARPP algorithm for all DEM resolutions is successful at

placing the peak power pixel close to the centre of the DDM window, significantly better than the quasi-spherical model. There is some noise in the results due to some DDMs presenting incoherent scattering as discussed in Section 4.3, however the overall impression is that the algorithm improves the placement of the peak.

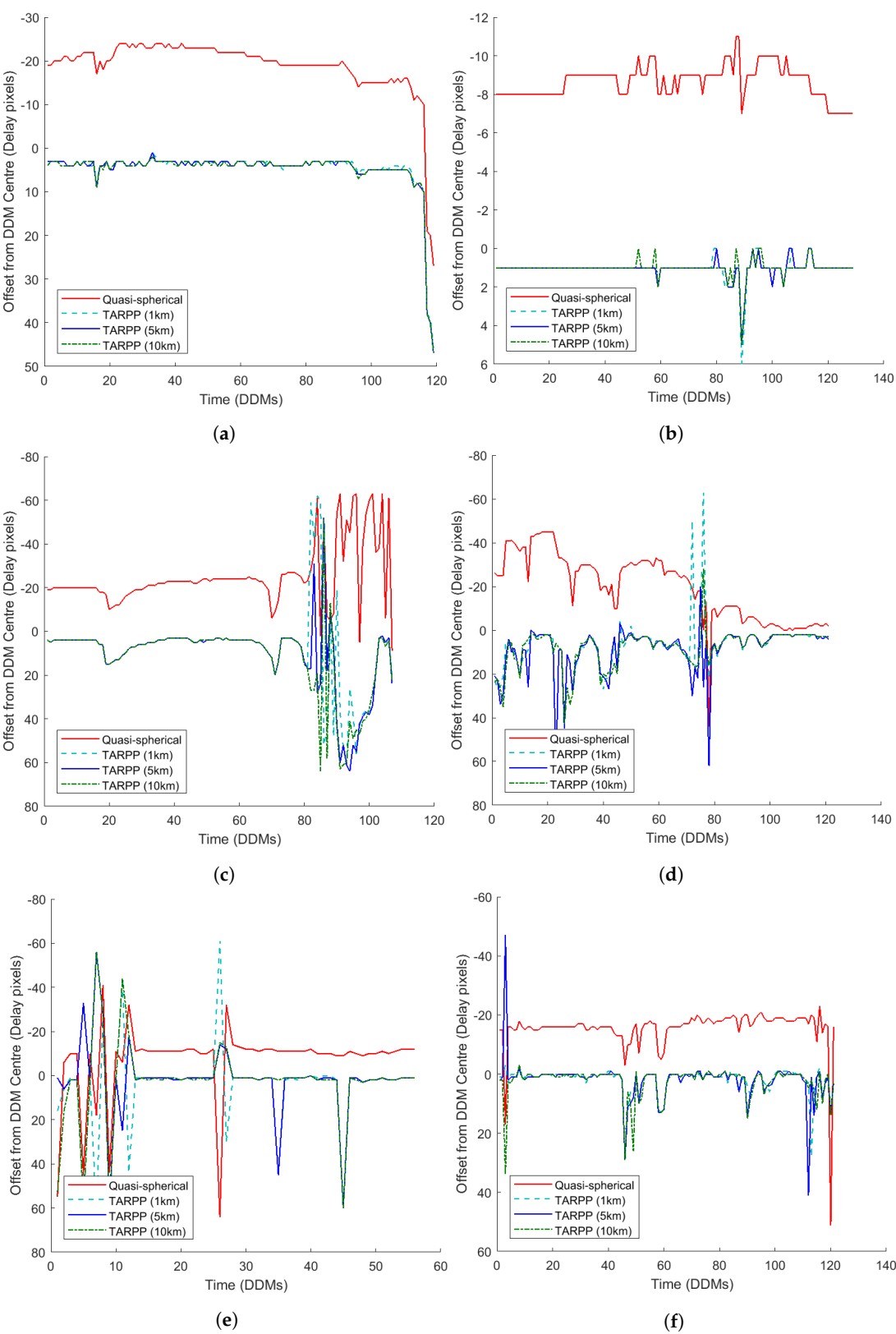

**Figure 5.** The results of Analysis 1a for an example PRN from each test dataset—plots of peak power offset from the centre of the DDM. (**a**) TT1, PRN 21; (**b**) TT2, PRN 15; (**c**) TT3, PRN 05; (**d**) TT4, PRN 20; (**e**) TT5, PRN 08; (**f**) TT6, PRN 11.

There are several points of interest: firstly, in many of the datasets there appears to be a constant bias in the location of the peak pixel below the centre of the DDM. Whilst this is preferable to a bias above the DDM centre, which could lead to contamination of the noise box, the cause should still be discovered and mitigated. The exact value of the bias is not common to all datasets and PRNs and so is unlikely to be a systematic, instrument-based bias however this cannot be said for sure without further tests. As discussed in Section 3.2, a similar bias found in another dataset was predicted to be caused by geoid undulations. However, analysis found that this was not the case, and so it is not predicted to be the reason here. Future analysis will investigate this possibility more thoroughly.

Secondly, taking TT1 as an example, there is a feature present in the data from all three resolutions towards the end of the DDM track which shows the peak power pixel shifting smoothly to greater delay values. This behaviour could be explained by a gradual decrease in elevation along the track of the specular point which is not accounted for by the DEM, however this is not corroborated by elevation maps of the area. Both these phenomena will be subject to further investigation.

Thirdly, in some instances, such as in the "Time (DDMs) > 80" portion of Figure 5c, the errors of the quasi-spherical and TARPP series appear to trend in opposite senses. In this case, this is believed to be due to very high elevations towards the end of the track (it ends at ∼5500 m). This would cause the peak reflected power to fall a large number of delay pixels outside the DDM—corresponding visually to the space "above" the DDM. The extended "arms" of the DDM horseshoe (see Figure 1a), which although lower in power than the peak are still higher than the noise background, could then contaminate the top portion of the DDM. This could cause the peak value to be recorded at low delay pixel values more often—corresponding to the top half of the graphs in Figure 5. Conversely, the TARPP method corrects the peak pixel on or close to the centre of the DDM, leaving the top part of the DDMs uncontaminated, other than the effects of incoherent scattering and noise. The simplifying assumptions of the TARPP v1 algorithm, which will be addressed in further developments, do not account for the possibilities of multiple reflection points, which may be contaminating the lower portion of the DDMs following application of the TARPP algorithm.

### 5.2. Analysis 1b

Analysis 1b was conducted on the six test datasets both individually and as a consolidated set, however for conciseness here only the consolidated results are presented. Figure 6 contains histograms describing the position of the peak power pixel relative to the centre of the DDM for a combination of the data from all six datasets. Figure 6a–c show that the TARPP algorithm, for all resolutions of DEM, is much better at placing the peak power pixel at, or close to, the centre of the DDM window than the quasi-spherical model. Figure 6d shows that the performance of the three DEM resolutions is very similar, with the 5 km resolution showing a very slight advantage over the others. The distribution is skewed to the right meaning it is less likely that there will be contamination of the noise box even if the peak pixel is not placed exactly at the centre.

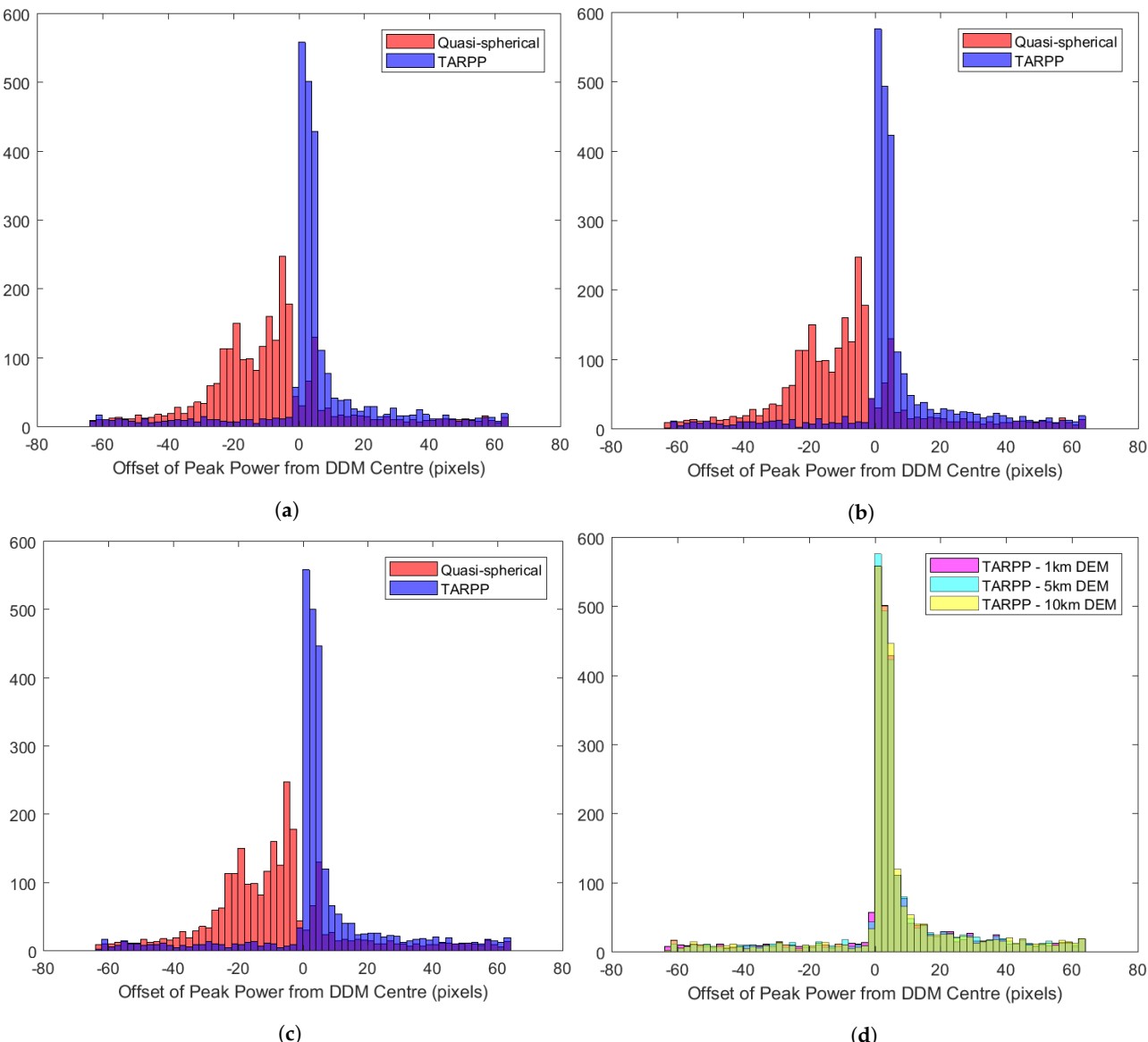

**Figure 6.** Histograms, generated using the combined data of all DDMs from all six datasets, showing the offset of the peak power pixel from the centre of the DDM. The three DEM resolutions are shown and compared with the quasi-spherical model in (**a**–**d**) shows the three DEM resolutions directly compared to each other. All three resolutions have a similar performance and thus the histograms nearly overlay each other, and so appear green. (**a**) 1 km DEM; (**b**) 5 km DEM; (**c**) 10 km DEM; (**d**) DEM Comparison.

### 5.3. Analysis 2

Analysis 2 tested how often the peak pixel was located within various sized boxes around the centre of the DDM, and was conducted on each test dataset individually and a consolidated set. Table 3 contains the results of Analysis 2 on the TT1 dataset, which is provided as an example to show the significant variation between the PRNs (which have different tracks on the Earth's surface). Reasons for the differences may include different surface roughness and different elevation angles to each of the transmitting satellites. In addition, as discussed in Section 4.3, many of the DDMs in all of the test datasets had a noise-like quality. This is due to a weak coherent scattering component, which can be caused by surface roughness or attenuation by vegetation. This could account for some of the differences between the PRNs, and is also a contributing reason to why the scores are lower than might be expected when considered in absolute terms, particularly when compared with the histograms in Figure 6 for example. However, for the purposes of this

study, a comparison is sufficient to demonstrate the improvement offered by the TARPP algorithm. In future work, a robust method for cleaning the datasets to leave coherent DDMs, which will lend themselves better to testing using these methods, will be developed.

**Table 3.** Results of Analysis 2 on the TT1 dataset.

| Dataset and PRN | No. of DDMs | Box Size (Pixels) | Score (% Success Rate of Placing Pixel in Box) | | | |
|---|---|---|---|---|---|---|
| | | | Quasi-Spherical | TARRP—1 km | TARPP—5 km | TARPP—10 km |
| TT1, 15 | 126 | 4 × 4 | 1.6 | 4.8 | 4.8 | 4.8 |
| | | 8 × 8 | 23.8 | 68.3 | 72.2 | 74.6 |
| | | 20 × 20 | 88.9 | 85.7 | 85.7 | 85.7 |
| TT1, 18 | 126 | 4 × 4 | 0 | 5.6 | 11.9 | 8.7 |
| | | 8 × 8 | 1.6 | 55.6 | 54.0 | 51.6 |
| | | 20 × 20 | 4.8 | 81.7 | 81.0 | 79.4 |
| TT1, 21 | 126 | 4 × 4 | 0 | 1.6 | 0.8 | 0.8 |
| | | 8 × 8 | 2.4 | 77.0 | 72.2 | 72.2 |
| | | 20 × 20 | 4.8 | 92.9 | 92.9 | 92.9 |

Table 4 shows the results of Analysis 2 with the data from all the datasets and PRNs combined using a weighted average, taking into account that there were slightly different numbers of DDMs for each dataset and PRN (due to different track lengths). This corroborates the results shown in Figure 6—that the three DEM resolutions each show very similar performances, with the 5 km resolution showing a very slight advantage (though this could be negligible when noise is considered). The 8 × 8 box size, which could theoretically allow 24/7 operations on DoT-1 through DDM windowing and thus data compression, has been tested. The results show that with the current test method there is a five-fold improvement over the quasi-spherical method. A 55% success rate of capturing the peak reflected power with this window size is not adequate for a fully operational service however it is a significant step closer than using the current methods. It should be noted that the lack of a 100% score, for example, does not necessarily mean that the algorithm stops working at certain points in the track, but rather that the signal may have been attenuated as described above and thus the maximum power no longer comes from the specular point pixel. More advanced test methods to be undertaken in future work should give a better indication of success rate.

**Table 4.** Results of Analysis 2 on all datasets, combined as a weighted average using number of DDMs for each dataset and PRN.

| Box Size (Pixels) | Weighted Average Score (% Success Rate of Placing Pixel in Box) | | | |
|---|---|---|---|---|
| | Quasi-Spherical | TARRP—1 km | TARPP—5 km | TARPP—10 km |
| 4 × 4 | 1.05 | 30.18 | 30.77 | 30.36 |
| 8 × 8 | 10.35 | 55.52 | 55.09 | 54.46 |
| 20 × 20 | 35.20 | 67.72 | 67.76 | 67.19 |

None of the three DEM resolutions shows a significant improvement over the others. It was expected that the higher the resolution, the better the performance, however this has not been demonstrated in the results. This may be explained by incoherent scattering dominating in the test datasets, causing the spatial resolution of the scattering to be larger than all three of the DEM resolutions, so that there is not significant distinction between them.

## 6. Future Work

This study has shown that although the baseline algorithm TARPP v1 is simple, it is able to improve the placement of the peak power pixel in the DDM window over

existing methods. However, it makes the assumption that the land surface is flat (as well as elevated), when this is not very often the case. Therefore development has now begun on the next stage of the algorithm, TARPP v2.

The TARPP v2 algorithm will include surface slopes to more accurately assess the equality of incidence and reflection angles, a criteria for determining the specular reflection point. Two components of slope from DEM data will be used—slope data products are typically provided along N-S and E-W lines, however the slope that is important to the reflection process is the slope projected in the incidence plane, defined for bistatic radar as the plane containing the transmitter, receiver and specular point positions (providing a smooth-Earth approximation is used). Complications arise when it is considered that the effect of topography may now mean that the shortest path to the ground is not the one that lays in the nominal incidence plane, and this will be considered during the development of the algorithm.

A trade-off will be conducted as to whether the slope components will be stored on-board, or calculated in real-time from the existing elevation values. Two components of slope plus elevation at every point would result in a trebling of the required data, affecting uplink times and storage, if the storage option is chosen. There will also be consideration of multiple reflection points, which can occur in regions of mountainous terrain or near cliffs.

It is expected that the TARPP v2 algorithm will show improved results compared with TARPP v1, including demonstrating the predicted improvement in performance with resolution (to a point—once the resolution is comparable to the best-possible spatial resolution of reflectometry, i.e., 1 km, it is unlikely that further improvements will be gained by increasing the resolution). As the envelope of GNSS reflectometry technology continues to be pushed and the use cases expanded, there will be a need for even more accurate specular point prediction. Therefore incorporated into the future work on the TARPP algorithm will be a consideration of the error terms introduced by the initial smooth-Earth estimate, including a comparison with the ellipsoidal Earth model.

## 7. Conclusions

This paper has presented the motivations for sensing land parameters with GNSS-R and demonstrated that a reflection point prediction method which takes topography into account is required to enable this. The TARPP v1 algorithm, which has been developed to meet this need on-board operational GNSS-R satellites, has been described along with planned future developments. Testing of this algorithm has been conducted using raw data from TDS-1 in a software receiver, and three horizontal resolutions of DEM have been tested with the algorithm −1 km, 5 km and 10 km.

The results show that the TARPP v1 algorithm performs better than the quasi-spherical model at centring the peak reflected power in a DDM, meeting the operational requirements significantly more often. It has also been shown to be suitable for operational use on-board a small reflectometry satellite. An interesting result is that no one of the DEM resolutions demonstrated significantly better results than the others, which is possibly explained by the largely incoherent nature of land surface scattering. The flight stage of testing will proceed with the 5 km resolution DEM, which has been shown to fit on-board the DoT-1 satellite. TARPP v1 has been written into C code ready for deployment to the satellite at a suitable time and testing will be undertaken in the near future.

Demonstrating that a topography algorithm can be used successfully on-board an operational spacecraft is a significant step towards 24/7 land sensing operations and the deployment of GNSS-R missions targeting Essential Climate Variables. This will help to achieve the GCOS goals of reducing satellite-sourced ECV data gaps.

**Author Contributions:** Conceptualization, L.K., M.U. and J.R.; methodology, L.K.; software, L.K., J.R., and M.U.; validation, L.K.; formal analysis, L.K.; investigation, L.K.; resources, M.U., J.R. and L.K.; data curation, M.U., J.R. and L.K.; writing—original draft preparation, L.K.; writing—review and editing, R.G., M.U., C.U. and L.K.; visualization, L.K.; supervision, C.U., M.U. and R.G.; project

administration, L.K.; funding acquisition, R.G., C.U., M.U. and L.K. All authors have read and agreed to the published version of the manuscript.

**Funding:** This research was funded by the Natural Environment Research Council via Lucinda King's SCENARIO PhD studentship, grant number NE/L002566/1. The PhD stipend was topped-up by Surrey Satellite Technology Ltd. The APC was funded by the University of Surrey.

**Data Availability Statement:** The data presented in this study are available in Figshare at https://doi.org/10.6084/m9.figshare.14171153.v1. They are available under a CC BY 4.0 license, and if used should be attributed to SSTL and Lucinda King.

**Acknowledgments:** Lucinda King would like to thank Surrey Satellite Technology Ltd. for providing the raw TDS-1 data used in this research.

**Conflicts of Interest:** The authors state that funding comes in part from Surrey Satellite Technology Ltd. (SSTL), owner of the TDS-1 and DoT-1 satellites and co-sponsor of the PhD study. SSTL offered support in design, collection, analysis and interpretation of data. The majority of PhD funding is from the Natural Environment Research Council who played no role in the design of the study; in the collection, analyses, or interpretation of data; in the writing of the manuscript, or in the decision to publish the results.

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
