# Peer review of "Towards a Topographically-Accurate Reflection Point Prediction Algorithm for Operational Spaceborne GNSS Reflectometry—Development and Verificationâ€"

_remotesensing, doi:10.3390/rs13051031_

Round 1
Reviewer 1 Report
In this manuscript, the Authors propose an algorithm for the estimation of the specular points of GNSS-R geometry. The paper is well written a well organized, even if I've found it a bit long. It can be made more concise, in my opinion.
The topic is of great interest for the community; a lot of researches are in progress to understand and describe the potential and the performance of the GNSS-R technology. The scientific background of the idea is clear.
I would like to propose some revisions in the following, which I believe can further improve the manuscript and the reader understanding. I believe they should be addressed before recommending the manuscript for publication.
- the way fig. 2 is generated would suggest that the specular point is the same in the presence of topography. Perhaps this is not the most challenging case. What about if the point under consideration is different?
- It is no clear which DEM is actually used. An SRTM or TandemX?
- I am not convinced about the resolution mentioned over land. 1 km is too optimistic, and it can be achieved over water bodies only. The scattering is mainly incoherent even over land, excluding a few particular cases. This is, possibly, one of the reasons why the Authors do not observe performance differences changing the DEM resolution. I agree that the DEM resolution should be smaller than the resolution to appreciate an improvement.
- It is very important, in my opinion, to specify the land type/cover for the regions under analysis. This would allow for better understanding the performance.
- It would be also nice and helpful to give a look at the DEM local slope for the region under analysis. In the presence of rough topography, I would expect that the DEM resolution can have a important role.
- How can the algorithm perform in the presence of vegetation? I agree that the vegetation attenuates the signal, but DEM models are not accurate under dense vegetation. The information uploaded on board could be not enough accurate. Note that the vegetated areas are of great interest for the applications.
- I would make clearer, if possible, how the specular points are currently estimated (I mean, for the dataset currently disseminated online, I understand with the quasi-spherical method but I am not sure). To my understanding with a large Doppler-delay window a very accurate estimation is not needed.
Reviewer 2 Report
General opinion:
This paper presents a method for the prediction of the positions of the reflection points in a space-based GNSS-R system. The authors present and evaluate the TARPP v1 algorithm, that applies a height correction, based on the EarthEnv Digital Elevation Model, to the reflection point position that is currently obtained without taking into account the topography of the scanned area. This algorithm is tested using TechDemoSat-1 real data. The algorithm generally performs better than the existing quasi-spherical approach. The paper is well-written and the overall aim of the paper is clear and can be interesting to many readers.
However, the work presented is at an early stage for a full paper publication, as indicated by the numerous paragraphs ending with a mention that there will be further work to address known issues. In particular, the assumption that the latitude and longitude of the reflection point are not dependent on the elevation (above the ellipsoid) at the reflection point is very rarely true, as indicated in the TARPP v2 section. This version of the algorithm is only briefly presented in the paper but would the version to complete and evaluate for full paper publication.
Also, several important statements throughout the paper are unclear and should be explained more precisely, as listed below:
- I am not used to TDS-1 data and I think that the paper is not clear enough in detailing which type of data is available onboard the satellite and on the ground and what are the limitations motivating the work. Section 2 indicates that the TDS-1 Level 1b DDMs (I guess this is the data available onboard the satellite?) are 128x20 pixels (what are the reasons for this limitation?) and the downlinked data should be an 8x8 pixels box with the same center. The figures in section 2 present cases where the reflection point to observe is not in the 8x8 pixels box but is inside the 128x20 box, making it seem like this is the problem to address. But then, the same section states that for elevations above the ellipsoid that are greater than 2300m, the error in the modeled path length is too large for the 128x20 size limitation. This is confusing: which limitation is actually motivating the work? Also, if the 128x20 Level 1b DDMs (or even larger) are available onboard, why not search for the position of the peak power pixel value in the Level 1b DDM and center the 8x8 pixels box around this position instead of using the center of the DDM?
- Table 1 lists the locations of the TDS-1 datasets used for the experimentations. There is no indication of the elevation above the ellipsoid for these locations, making it hard to evaluate how representative these locations are and how important the use of the TARPP algorithm for these datasets is.
- In section 4.3, the authors indicate that they mitigate the noise-only DDMs by only considering DDMs which have the peak power pixel within ±2 Doppler pixels of the center. This, or the use of 8x8 DDM pixel boxes, is questionable. If only 4 Doppler pixels are necessary to make sure the peak power pixel is present, why are 16x4 boxes not used instead of 8x8 boxes? Also, the reasons for taking only 4 Doppler pixels around the center of the DDM should be explained. Finally, why not use a threshold in the ratio between the peak power pixel value and the noise level to determine if the DDMs are or not noise-only, similar to what is usually done in the classic GNSS acquisition process?
- The paper does not provide any explanation or description on Table 2. Its importance and contribution to the work should be better clarified especially that it is included as a section by itself. Only the first column information, which is again defined later, seem to be useful as the other columns are very straightforward calculations results.
- The authors state that the DDMs sent to earth are 8x8 pixel boxes, however, in the experimentation section, the authors test the 4x4, 10x10, and 20x20 box pixel sizes. It is better if the authors apply the experimentations using the desired box size.
- In the experimentation section, the assessment of the proposed algorithm is primarily based on the analysis of the TT1 dataset, and the authors detail the results for PRN21 in particular. However, there is no clear explanation on why this dataset, PRN, and set of PRNs (15, 18, 21) were chosen specifically to show the main results of the work. Section 5.2 is the actual section containing the general results, but the size of the tables, figures, and explanatory comments make it seem like section 5.1 is the section containing the main results of the work.
- To end up, the paper states that preliminary work using Gaussian smoothing did not show any significant improvement to the results. What is the purpose of section 5.3? This point could have been just mentioned as preliminary work without showing the results in a complete section.
Reviewer 3 Report
see attached file

Round 2
Reviewer 2 Report
The authors have improved the article significantly, in particular in explaining the objectives and restrictions of their work and in the reorganization of the sections across the whole paper. The presentation of the results is much clearer as well. In my opinion, the paper in the present form can be considered for publication, but would still benefit from two additions:
- In Table 2, reporting the incidence angle and/or the path length difference could be useful.
- In section 5.1, a third point of discussion would be to try to explain what is going on when the TARPP errors become large in figure 5. I guess this is due to an overall increase in the elevation of the reflecting locations. I also suggest that a clear, precise motivation for TARPP v2 could be made from that point of discussion. If the authors have an explanation on how the TARPP errors can go very differently from the quasi spherical model errors (e.g at Time>80 in subplot (c), where the error variations go one way for TARPP and the other way for the QS model), it could make an interesting point of discussion as well.
Reviewer 3 Report
Good work. I am satisfied with the revisions made. The paper can be published as is.
Author Response
Many thanks for taking the time to review the paper a second time. We are pleased that you are satisfied with our revisions and are happy for the paper to be published in its current form.